# LLM4Cov: Execution-Aware Agentic Learning for High-coverage Testbench Generation

Hejia Zhang [1]  Zhongming Yu [1]  Chia-Tung Ho [2]  Haoxing Ren [3 †]  Brucek Khailany [2]  Jishen Zhao [1]

## Abstract

Execution-aware LLM agents offer a promising paradigm for learning from tool feedback, but such feedback can be expensive and slow to obtain, making online reinforcement learning (RL) less practical in certain scenarios. High-coverage hardware verification exemplifies this challenge due to its reliance on industrial simulators and non-differentiable execution signals. We propose LLM4Cov[1], an offline agent-learning framework that models verification as single-step state transitions guided by deterministic evaluators. Building on this formulation, we introduce execution-validated data curation, policy-aware agentic data synthesis, and worst-state-prioritized sampling to enable scalable learning under execution constraints. We further curate a reality-aligned benchmark adapted from an existing verification suite through a revised evaluation protocol. Using the proposed pipeline, a compact 4B-parameter model achieves 69.2% pass rate and 90.4% average coverage in CVDP-ECov under agentic evaluation, outperforming its teacher by 5.3% and 10.5%, demonstrating competitive performance against models an order of magnitude larger.

## 1. Introduction

Large language model (LLM) agents have demonstrated strong potential by interacting with tools and learning from execution feedback. This execution-aware paradigm is essential for agentic learning, as it grounds symbolic generation in real-world correctness through signals that cannot be inferred from text alone. However, training such agents

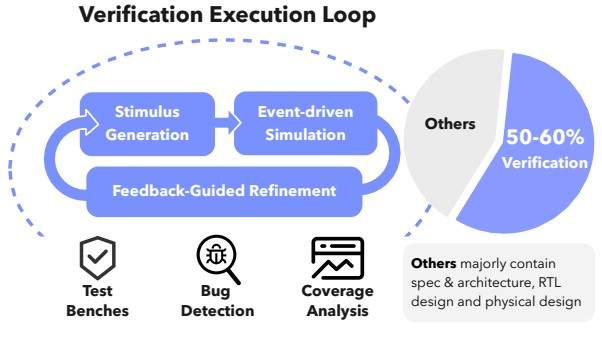

Figure 1. Execution-aware verification loop and its dominant cost in modern hardware design. (Hegde, 2025; Foster, 2025; 2017)

remains difficult: online learning can be less practical due to the massive runtime overhead and high cost of tool invocations, while the resulting execution traces can be challenging for standard fine-tuning objectives.

A critical yet underexplored domain for execution-aware agents is *hardware verification*. Before fabrication, a hardware design must be validated in simulation using a *testbench*—an executable verification program that generates input stimuli, drives the design, and measures coverage over signals, branches, etc. As shown in Figure 1, testbench-driven simulation and iterative refinement account for the majority of hardware design effort. (Hegde, 2025; Foster, 2025; 2017). Unlike software testing, hardware failures cannot be patched post-deployment and must be resolved under cycle-accurate execution semantics, making verification both execution-intensive and engineering-heavy. Following prior work (Pinckney et al., 2025; Nadimi et al., 2025), automated hardware verification can be broadly decomposed into two components: (i) *maximizing coverage through stimulus generation*, and (ii) *detecting bugs with assertions*. In this work, we focus exclusively on the former.

While coverage provides dense, comparable feedback, making it a natural verifier for iterative agentic refinement, it is non-trivial to turn this signal into effective supervision. Each evaluation requires expensive simulation, rendering large-scale online RL impractical and forcing the model to rely primarily on offline trajectories. Furthermore, using

---

[†]Work done while in NVIDIA. [1]University of California San Diego, USA [2]NVIDIA [3]Agentrys. Correspondence to: Jishen Zhao <jzhao@ucsd.edu>.

*Proceedings of the 43rd International Conference on Machine Learning*, Seoul, South Korea. PMLR 306, 2026. Copyright 2026 by the author(s).

[1]The open-source implementation of LLM4Cov is available at https://hejiaz2023.github.io/llm4cov_oss/.

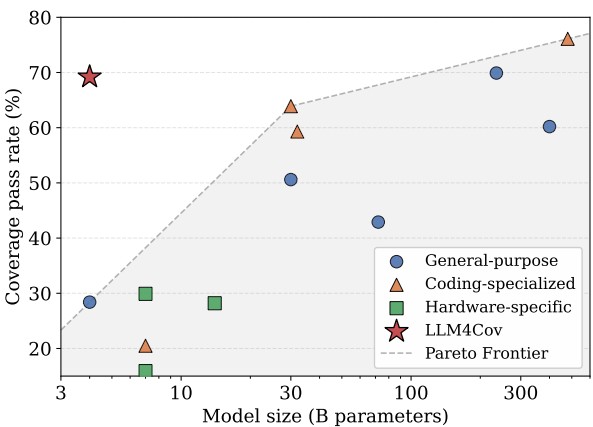

*Figure 2.* Coverage pass rates of existing LLMs. Results are measured in agentic setting on our benchmark (Section 4.1).

static datasets induces a state-dependent distribution shift: the intermediate failures encountered by a student model differ substantially from those in teacher-generated datasets. Existing approaches do not directly address this regime of dense but costly feedback under offline, distribution-shifting agentic learning. Consequently, existing approaches do not provide a systematic framework to extract maximal supervision from coverage signals while aligning training data with the evolving student model.

To address these challenges, we introduce LLM4COV, the first execution-aware agentic learning framework for high-coverage testbench generation that systematically converts coverage feedback into stable offline supervision. At its core is **Coverage-Guided Agentic Rejection Fine-tuning**, which treats coverage as a dense supervisory signal to filter and refine synthesized trajectories: testbench drafts are generated by the student model, and low-coverage drafts together with their most coverage-improving revisions are retained, concentrating supervision on recoveries and extracting maximal information from each simulator run. Because these trajectories depend on the current student, we further propose **Verification-Conditioned Progressive Learning**, in which synthetic data are generated in a staged manner and training also proceeds in stages aligned with the evolving student distribution; this progressive supervision yields significantly better final performance compared to naive data augmentation. As shown in Figure 2, we demonstrate that an execution-aware 4B-parameter model trained with our framework can outperform 30B-class teachers and demonstrate performance competitive with $50\times$–$100\times$ larger models, proving that specialized agentic learning can achieve high-coverage verification results with far greater efficiency than general-purpose scaling.

**Conflict of Interest Disclosure.** This work is supported by the NVIDIA Academic Grant Program, which provided A100 GPU via the Brev platform. Authors C.H., H.R. and B.K. are employed by NVIDIA during this work. Some of the benchmarks used in our evaluation (e.g., CVDP (Pinckney et al., 2025), VerilogEval (Liu et al., 2023)) were introduced in prior work co-authored by NVIDIA employees, including some authors of this paper. To the best of our knowledge, no model or baseline evaluated in this paper has NVIDIA affiliation.

## 2. Background and Related Work

### 2.1. Execution-Aware Agent Learning

**Imitation learning and state-distribution alignment.** In imitation learning, DAgger mitigates covariate shift by aggregating expert labels on states visited by the current policy (Ross et al., 2011). Recent work extends this insight to language-model agents: Fine-Tuning Web Agents identifies off-policy bias as a central failure mode of expert-trajectory SFT and emphasizes the importance of training on student-induced states (Caccia et al., 2024). On-policy Expert Corrections (OEC) further mitigates this by switching from student to expert guidance mid-rollout, producing partially on-policy supervision (Lauffer et al., 2025).

**Trajectory filtering and failure-aware supervision.** Exploring Expert Failures shows that discarding unsuccessful trajectories overrepresents easy cases and that incorporating failure segments improves agent tuning (Lan et al., 2025). STeCa identifies suboptimal actions via step-level reward comparison and constructs calibrated trajectories through online exploration (Wang et al., 2025a). Agent-R similarly relies on iterative self-training with online rollouts and search-based trajectory revision (Yuan et al., 2025). While effective, they require online interactive environments and trajectory-level optimization.

**Progressive and multi-stage learning.** Progressive Distillation shows that gradually distilling from stronger teachers can induce an implicit curriculum, improving stability and final performance even without explicit curriculum design (Panigrahi et al., 2025). In a complementary direction, multi-stage fine-tuning in continual learning settings demonstrates that organizing supervision across stages can mitigate interference and improve adaptation when models are updated sequentially (Guan et al., 2025). These works highlight the value of stage-structured supervision, but generally assume changing teachers or continual data updates. They do not address the regime where task and teacher are fixed while the student's state distribution evolves through execution, nor how to construct stage-conditioned supervision from expensive offline feedback.

**Offline Agentic learning.** Prior work on offline LLM agent learning has primarily focused on (i) demonstrating feasibility of trajectory-level SFT at scale (Song et al., 2024), (ii) assigning value to intermediate steps or actions in long-

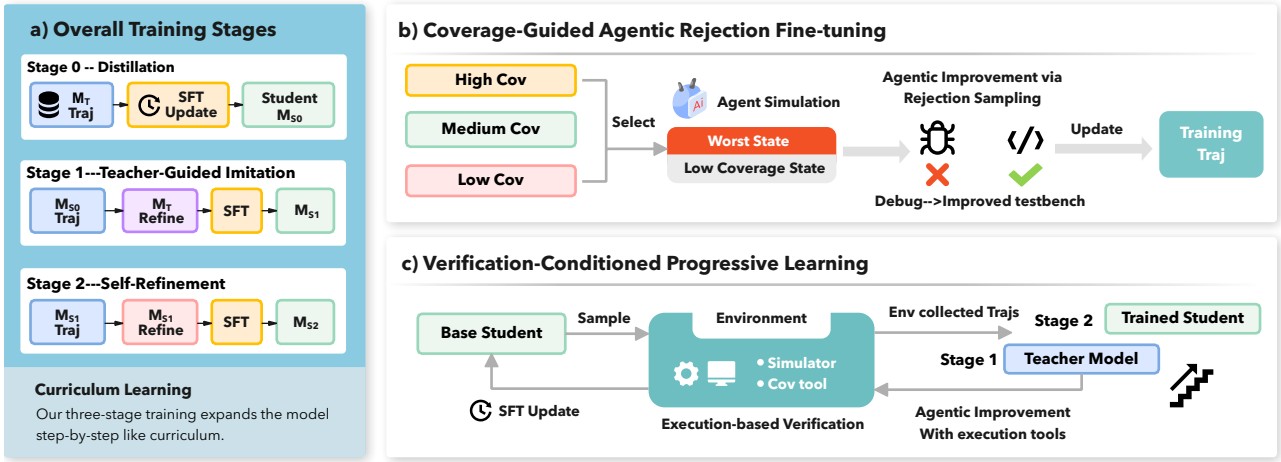

*Figure 3.* Main components of LLM4COV. (a) The framework samples trajectories to SFT the student model, and the process is separated into stages to align with the evolving student distribution. (b) Coverage-Guided Agentic Rejection Fine-tuning retains low-coverage drafts and their most coverage-improving revisions, concentrating supervision on recovery behaviors. (c) Verification-Conditioned Progressive Learning generates and trains on staged synthetic trajectories conditioned on the current student, yielding progressively stronger agentic performance and more stable final coverage.

horizon agent trajectories (Deng et al., 2025; Lin et al., 2025), and (iii) scoring entire trajectories for data selection (Zhang et al., 2025).

In contrast, our work targets a different and underexplored regime: single-step, iterative refinement agents with existent dense, ground-truth scoring (coverage) for each step. This setting departs from multi-step credit assignment and instead raises a distinct optimization problem—how to construct and select training data when each transition is directly verifiable and comparable. Our contributions on Coverage-Guided Agentic Rejection Fine-tuning are designed specifically for this regime, serving as an execution-aware agent learning framework enabling effective offline learning.

## 2.2. LLMs for Hardware Design and Verification

**Background on hardware verification.** Hardware verification validates design correctness before fabrication by executing the design under *testbenches*, which specify input stimuli and expected behaviors. These testbenches are evaluated using cycle-accurate *simulators* that model the hardware's execution semantics. Verification progress is measured through *coverage* metrics, which quantify how thoroughly the design's logic and behaviors are exercised.

**Recent work on LLM for hardware.** Several recent studies applied LLMs to hardware design and verification tasks. For hardware design, prior efforts primarily target generating hardware description languages (HDLs) from natural-language specifications, improving model reasoning or training objectives for hardware synthesis through structured datasets or RL with verifiable reward (Wei et al., 2025; Zhu et al., 2025; Wang et al., 2025b). Complementary agent-

based systems further decompose hardware code generation into multi-step workflows with tool interaction (Zhao et al., 2025b; Ho et al., 2025). A smaller body of work studied LLMs for hardware verification, focusing on generating hardware testbenches and verification stimuli that exercise design behaviors under simulation. Existing systems adopt iterative generation and validation pipelines using tool feedback from simulator execution and rule-based checking (Qiu et al., 2025; Zhao et al., 2025a). Recent work further applies offline preference optimization to testbench generation by constructing coverage-labeled preference pairs and training via DPO, with preference strength scaled by coverage gaps between candidates (Nadimi et al., 2025). While effective for single-shot stimulus generation, these approaches do not study interactive repair, state-distribution shift, or agentic learning under tool feedback.

In contrast, our work models verification as a sequence of evaluator-scored state transitions and explicitly addresses multi-turn interaction and distribution alignment under limited simulator execution budgets.

## 3. Methodology

Our framework (Figure 3) converts expensive simulator feedback into stable offline supervision for training verification agents. We first formalize the verification setting and the supervision signals available from simulator execution. Section 3.1 describes the feedback returned by the simulator and the coverage metric used to measure progress. Section 3.2 models verification as a sequence of single-step state transitions and defines the transition data points used for training. Sections 3.3 and 3.4 then describe how these

data points are constructed from agentic tarjectories and organized across training stages to align supervision with the evolving student model.

### 3.1. Simulation Feedback and Coverage

We employ a simulator as an evaluator that executes testbenches against a fixed hardware design repository (which consists of the design source files and specifications) and returns structured feedback. Given a generated testbench, a simulator invocation returns a feedback observation tuple: $o_{feedback} = (\texttt{status}, \texttt{coverage}, \texttt{log})$, where

- $\texttt{status}$ indicates the execution outcome, including compile failure, runtime failure, successful completion, etc.
- $\texttt{coverage}$ reports quantitative coverage metrics collected during simulation;
- $\texttt{log}$ contains diagnostic information such as compiler errors, assertion failures, or runtime traces.

We aggregate simulator-reported coverage metrics into a normalized scalar score via a coverage function:

$$\text{Cov}(\cdot) : o_{feedback} \to [0, 1].$$

Simulator calls can dominate computational cost. In academic settings, a single execution typically requires seconds, while industrial flows may require minutes or hours. To enable fair comparison, we fix simulator call budgets across all ablation studies.

### 3.2. Agentic Verification as Single-Step State Transition

We formalize execution-aware verification as an iterative generation process in which a language model repeatedly produces verification programs and a simulator deterministically evaluates them. Our central assumption is transition being *single-step*: all information available to the agent is explicitly represented in the current state.

**State.** Let $\mathcal{R}$ denote a fixed hardware design repository, consisting of all design source files and specifications. At step $t$, the state and its coverage are defined as

$$s_t = (\mathcal{R}, x_t, o_t), \quad \text{Cov}(s_t) \triangleq \text{Cov}(o_t) \in [0, 1],$$

where $x_t$ is the *full testbench file* at step $t$, and $o_t = (\texttt{status}, \texttt{coverage}, \texttt{log})$ is the simulator feedback observation defined in Section 3.1. We define the initial state as $s_0 = (\mathcal{R}, x_0, o_0)$, where $x_0 = \varnothing$ denotes an empty testbench placeholder and $o_0 = \varnothing$ represents a null simulator observation prior to any execution. For the initial state, we define $\text{Cov}(o_0) = 0$.

Although $\mathcal{R}$ is constant across transitions, the full repository contents are provided to the model at every step. We focus on full testbench regeneration rather than patch-based editing, as fine-grained localization and diff generation are unreliable without specialized pretraining on repository-level

edit traces, and introducing partial edits would confound the learning objective with additional structural assumptions.

**Transition.** A single transition composes LLM inference with simulator execution:

$$\begin{aligned} x_{t+1} &\sim M_\theta(\cdot \mid s_t), \\ o_{t+1} &= \text{Sim}(\mathcal{R}, x_{t+1}), \\ s_{t+1} &= (\mathcal{R}, x_{t+1}, o_{t+1}) = (\mathcal{R}, x_{t+1}, \text{Sim}(\mathcal{R}, x_{t+1})). \end{aligned}$$

**Single-step assumption.** The LLM inference depends only on the current state representation:

$$M_\theta(x_{t+1} \mid s_{0:t}) = M_\theta(x_{t+1} \mid s_t),$$

where any information from prior interaction rounds must be encoded explicitly in $(x_t, o_t)$ rather than implicit history. This formulation preserves all information necessary for our specific task, which is improving verification coverage, since earlier trials are subsumed by the updated code and observation. Moreover, discarding raw interaction history reduces prompt length and redundancy, allowing the model to focus on the most recent execution signal.

To provide a diagnostic comparison, we define a *vanilla agent* as one that conditions each generation on the full interaction history $(s_0, a_0, o_0, \ldots, s_t)$, rather than the memoryless state representation $s_t$. Table 1 shows that the memoryless formulation consistently yields equivalent or superior performance across both compact and larger models.

**Data point formulation.** We define a data point as $d_t = (s_t, x_{t+1})$, where $s_t = (\mathcal{R}, x_t, o_t)$ is the current state and $x_{t+1}$ is the verification program generated from that state. Learning therefore reduces to constructing a dataset of transitions $d_t = (s_t, x_{t+1})$ that improve verification coverage under simulator feedback.

**Difference from standard Markov modeling.** Markov formulations are standard in RL, and that non-Markovian problems can be made Markovian by including full history. However, our formulation on agent learning can operate with a single-step, history-minimal state. Recent work and reports (Zhang et al., 2024; Sun et al., 2025; Hong et al., 2025) highlight degradation in long-context settings and motivate history compression or decomposition. Our approach can be viewed as a principled extreme of this trend: instead of compressing history, we minimize it and rely on the current execution state.

### 3.3. Coverage-Guided Agentic Rejection Fine-Tuning

Section 3.2 defines verification as a sequence of state transitions evaluated by a simulator. Learning therefore reduces to constructing supervision pairs $d_t = (s_t, x_{t+1})$ that improve coverage under execution feedback. We now describe how

*Table 1.* Effect of the single-step formulation on agents. Results are measured on our verification benchmark (Section 4.1).

| | Pass@1 (higher is better) | |
| --- | --- | --- |
| Model | Vanilla | Single-Step |
| Qwen3-Coder-30B-A3B-Instruct | 59.5% | **63.9%** |
| Qwen3-4B-Instruct-2507 | 28.2% | **28.4%** |

such data points are synthesized from agentic trajectories and filtered using coverage signals.

**Trajectory synthesis.** Under the memoryless transition model, all trajectories originate from the same initial state $s_0 = (\mathcal{R}, \varnothing, \varnothing)$. Consequently, differences in agentic behavior arise not from the initial state itself, but from how *intermediate states* are sampled and how transitions are generated from those states.

We characterize an agentic trajectory by two models: (i) $M_{\text{int}}$ used to sample intermediate states, and (ii) $M_{\text{trans}}$ used to generate the final transition from those states. Starting from the initial state $s_0$, a trajectory of length $N$ is generated by iteratively sampling the intermediate-state model for the first $N-1$ rounds, followed by a final transition generated by the transition model,

This formulation allows the intermediate-state distribution to be controlled independently from the quality of the final transition, and naturally subsumes both imitation-style and self-sampling trajectories as special cases. For simplicity, we set $N = 2$ when synthesizing dataset in our experiments.

**Worst-state selection.** Uniformly generating transitions from all visited states tends to overrepresent already successful contexts and yields limited corrective supervision. Instead, we concentrate synthesis on failure-prone regions.

From the initial state $s_0$, we sample a set of candidate intermediate states $\mathcal{S}_{\text{cand}} = \{s^{(1)}, s^{(2)}, \dots\}$ using $M_{\text{int}}$. Each state has an associated coverage score $\text{Cov}(s^{(i)})$. We select the lowest-coverage state

$$s_{\text{worst}} = \arg\min_{s \in \mathcal{S}_{\text{cand}}} \text{Cov}(s)$$

and generate corrective transitions from this state. Focusing on the worst-performing state increases the probability that sampled transitions contain useful recovery behavior under a fixed simulator budget.

**Coverage-guided rejection.** For each selected state $s_t \in \mathcal{S}_{\text{worst}}$, we generate transition candidates using the transition model $M_{\text{trans}}$:

$$x_{t+1} \sim M_{\text{trans}}(\cdot \mid s_t), \quad o_{t+1} = \text{Sim}(\mathcal{R}, x_{t+1}).$$

Each resulting data point $d_t = (s_t, x_{t+1})$ is filtered through execution-based rejection sampling:

$$\mathcal{F}_{\text{stage}}(d_t) = \mathbb{I}\Big[\text{Cov}(o_{t+1}) - \text{Cov}(o_t) \geq \tau_\Delta\Big],$$

where $\tau_\Delta$ denotes a minimum coverage improvement threshold. Among retained candidates, we keep the transition with the largest coverage improvement. This rejection mechanism converts simulator feedback into a dense supervisory signal that prioritizes corrective behaviors over already successful cases.

**Resulting supervision.** Applying worst-state selection and coverage-based rejection yields a dataset of transitions concentrated on recovery from low-coverage states. These data points are then used for supervised fine-tuning of the student model. By grounding supervision in failure modes and filtering transitions by execution improvement, the procedure extracts maximal learning signal from each simulator call while remaining fully offline.

### 3.4. Verification-Conditioned Progressive Learning

Section 3.3 describes how coverage-guided rejection fine-tuning converts simulator feedback into supervision pairs $d_t = (s_t, x_{t+1})$. Under such formulation, agentic traces can be categorized into three types depending on model selection between teacher model $M_T$ and student model $M_S$ (which is shown in Figure 4):

- **Full-teacher agentic traces.** Both intermediate states and transitions are generated by the teacher model, i.e., $(M_{\text{int}}, M_{\text{trans}}) = (M_T, M_T)$. These traces provide high-quality transitions but induce a teacher-biased state distribution that may exclude failure modes commonly encountered by the student.
- **Imitation-style agentic traces.** Intermediate states are sampled using the student model, while transitions are generated by the teacher model, i.e., $(M_{\text{int}}, M_{\text{trans}}) = (M_S, M_T)$. This formulation aligns supervision with the student-induced state distribution while preserving expert-

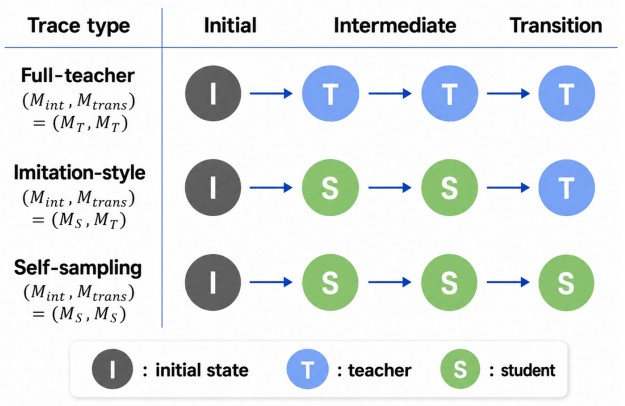

*Figure 4.* Student-grounded trajectory synthesis. Intermediate states are sampled by $M_{\text{int}}$, while the final transition by $M_{\text{trans}}$.

level corrective transitions.

- **Self-sampling agentic traces.** Both intermediate states and transitions are generated by the student model, i.e., $(M_{\text{int}}, M_{\text{trans}}) = (M_S, M_S)$. These traces fully reflect the student's execution behavior and remove reliance on a fixed teacher model.

**Training progression.** The above taxonomy decouples the distribution of visited states from the quality of corrective transitions and provides a simple progression for constructing supervision. When the student model is substantially weaker than the teacher, imitation-style traces offer stable learning signals by pairing student-induced failure states with strong corrective transitions from the teacher. As training proceeds and the student becomes capable of producing higher-quality repairs, self-sampling traces become increasingly valuable: they reflect the student's own execution behavior and enable learning recovery strategies for failure modes that lie beyond the fixed performance ceiling of a static teacher. In this way, trajectory synthesis can naturally evolve from teacher-guided correction toward student-driven refinement while remaining grounded in the same coverage-based filtering mechanism.

**Stage-conditioned data synthesis.** The progression above can be naturally implemented via separating data synthesis into stages. Let $M^{(k)}$ denote the student model at stage $k$. Using the procedure in Section 3.3, we construct a dataset $\mathcal{D}^{(k)}$ by sampling trajectories with $M^{(k)}$ and retaining data points containing coverage-improving transitions:

$$\mathcal{D}^{(k)} = \left\{ (s_t, x_{t+1}) \, \middle| \, x_{t+1} \sim M^{(k)}(\cdot \mid s_t), \, \mathcal{F}_{\text{stage}}(d_t) = 1 \right\}$$

Each stage therefore yields supervision aligned with the state distribution induced by the current student model.

**Staged supervised fine-tuning (SFT).** Given dataset $\mathcal{D}^{(k)}$, we update the student by standard supervised fine-tuning:

$$\theta^{(k+1)} = \arg \min_{\theta} \, \mathbb{E}_{(s,x) \sim \mathcal{D}^{(k)}} \left[ - \log M_{\theta}(x \mid s) \right].$$

Training proceeds sequentially across stages, $\theta^{(0)} \to$

$\theta^{(1)} \to \cdots \to \theta^{(K)}$, where each stage uses data synthesized from the previous checkpoint.

**Comparison to vanilla data augmentation.** As shown in Figure 5, a common alternative is to augment the old dataset with new synthesized data and train a single model on the union. However, this treats supervision from different student distributions as exchangeable. Because later-stage datasets contain more informative recovery behaviors for states encountered by stronger models, mixing all stages uniformly can dilute the training signal associated with the current execution regime.

**Progressive supervision.** In contrast, staged training preserves alignment between supervision and the evolving student model. Early stages emphasize syntactic validity and basic execution success through teacher-guided corrections. Later stages increasingly emphasize recovery from low-coverage states encountered during autonomous execution. This verification-conditioned progression stabilizes training and improves final coverage by ensuring that supervision remains concentrated on the failure modes most relevant to the current model.

## 4. Experiments

### 4.1. Benchmark and Metrics

**Benchmark.** We evaluate our method on 2 benchmarks: **CVDP-ECov.** We selected task `cid012` from the CVDP benchmark (Pinckney et al., 2025), which contains 83 independent hardware repositories. In the original setting, a testbench is generated solely from a natural-language design specification and evaluated based on whether its achieved coverage exceeds a specialist-defined threshold. In our adaptation, only the evaluation protocol differs. Specifically, the full hardware repository is made visible to the LLM rather than specification only. This protocol better reflects practical verification workflows that rely on both coverage feedback and hardware code. **AutoEval-ECov.** Following Correct-Bench (Qiu et al., 2025), we transformed VerilogEval (Liu et al., 2023) into a testbench coverage optimization benchmark by offering golden RTL and problem specification as input. As a result, we got 156 more tasks to be evaluated on, with same format as CVDP-ECov.

**Evaluation metrics.** We report three metrics:

- **Pass Rate**: percentage of repositories where coverage exceeds the predefined threshold; The threshold was defined by human experts in CVDP-ECov and was set to 100% in AutoEval-ECov.
- **Avg Cov**: average coverage across all repositories, assigning 0% where simulation fails.

Unless otherwise specified, all results are reported using **Pass@1** with $n = 5$ samples, aligning with CVDP.

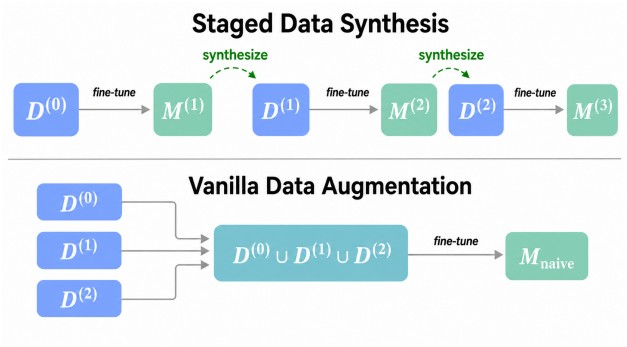

*Figure 5.* Staged data synthesis v.s. vanilla augmentation. Stage-specific supervision align with current student states, while mixing data together may dilute later stage signals.

*Table 2.* Evaluation results on CVDP-ECov and AutoEval-ECov benchmarks. Best metrics are **bolded** and second runners are underscored. Following the original CVDP evaluation, we treat coverage pass rate as the primary metric. Because all training stages optimize agentic execution—the primary metric and training objective—direct-inference results are reported only as a reference and may slightly degrade in later stages. All models evaluated in instruct mode and recommended config (reported in Appendix B). CorrectBench Baseline is an exception as it is a mult-agent framework by itself and therefore has no Direct Inference result. As Claude 3.5 Sonnet is no long provided by Anthropic, the original authors kindly sent us their evaluation results which we calculated AutoEval-ECov metrics upon. See Appendix D for more details on baseline comparisons.

| | | CVDP-ECov | | | | AutoEval-ECov | | | |
|---|---|---|---|---|---|---|---|---|---|
| | | Agentic | | Direct Inference | | Agentic | | Direct Inference | |
| Type | Model | **Pass %** | Avg Cov | Pass % | Avg Cov | **Pass %** | Avg Cov | Pass % | Avg Cov |
| General Purpose | llama-4-maverick (400B) | 60.2% | 81.7% | 23.1% | 47.3% | 32.9% | 48.7% | 21.4% | 41.9% |
| | Qwen3-235b-A22b-2507 | **69.9%** | 83.2% | 29.9% | 47.0% | 84.0% | 92.6% | 70.6% | 83.4% |
| | Qwen2.5-72B | 42.9% | 64.1% | 16.4% | 32.2% | 63.8% | 87.6% | 20.9% | 73.3% |
| | Qwen3-30B-A3B-2507 | 50.6% | 68.0% | 30.1% | 52.2% | 81.9% | 91.6% | 72.3% | 86.1% |
| Coding Specialized | Qwen2.5-Coder-32B | 59.3% | 79.6% | 23.4% | 52.1% | 74.5% | 88.0% | 52.1% | 80.1% |
| | Qwen3-Coder-30B-A3B | 63.9% | 79.9% | 28.4% | 48.6% | 83.8% | 94.0% | 75.3% | 87.1% |
| | Qwen2.5-Coder-7B | 20.5% | 34.4% | 11.3% | 26.6% | 59.9% | 77.5% | 20.9% | 52.2% |
| Hardware Specific | VeriCoder-Qwen2.5-14B | 28.2% | 47.9% | 17.1% | 37.3% | 54.0% | 69.2% | 28.3% | 56.2% |
| | CodeV-R1-RL-Qwen-7B | 29.9% | 55.5% | 14.5% | 32.5% | 68.5% | 85.6% | 42.6% | 67.9% |
| | VeriReason-Qwen2.5-7B | 15.9% | 26.4% | 8.9% | 16.2% | 41.5% | 55.0% | 13.7% | 27.9% |
| CorrectBench Baseline | Claude 3.5 Sonnet | N/A | N/A | - | - | 61.5% | 89.9% | - | - |
| | Qwen3-235b-A22b-2507 | 34.9% | 60.5% | - | - | 55.8% | 84.9% | - | - |
| LLM4Cov | Qwen3-4B-2507 (Base) | 28.4% | 48.5% | 14.9% | 23.3% | 48.3% | 62.8% | 31.0% | 50.9% |
| | +Stage0 | 60.2% | 82.1% | 45.5% | 74.9% | 82.7% | 94.7% | 78.5% | 92.0% |
| | +Stage1 | 67.0% | 88.2% | 48.2% | **77.2%** | 83.6% | 95.5% | 79.5% | **93.8%** |
| | +Stage2 | 69.2% | **90.4%** | **50.6%** | 76.4% | **85.0%** | **96.3%** | **79.6%** | 92.6% |

**Evaluation settings.** We evaluate model performance under two settings. *Agentic* evaluation (our primary setting) allows the model to iteratively generate and refine a testbench using simulator feedback for $N = 3$ rounds, with the single-step assumption from Section 3.2; *Direct Inference* evaluation (our reference setting) measures single-pass generation without any refinement or feedback.

### 4.2. Experimental Setup

**Models.** We obtain our final model by fine-tuning Qwen3-4B-Instruct-2507 on synthetic data generated by a teacher model (Qwen3-Coder-30B-A3B-Instruct) and itself.

**Dataset.** To generate synthetic testbench data, we used the hardware repo dataset from CodeV-R1 (Zhu et al., 2025), which contains 87k independent repos. To prevent benchmark contamination, we remove repositories that contain at least one file similar to any file in the CVDP-ECov benchmark, thresholding with 50% ROUGE-L similarity (CodeV-R1 already did same deduplication on VerilogEval, which is transformed into our AutoEval-ECov benchmark). See Appendix B for detailed SFT and EDA settings.

**Multi-stage progressive learning.** We adopt a 3-stage supervised fine-tuning (SFT) pipeline that progressively aligns the model with increasingly challenging agentic behaviors. Stage 0 training uses a warmup dataset where supervision

is generated from full-teacher agentic traces. Coverage-guided rejection similar with Section 3.3 is also applied, while an additional minimal coverage percentage requirement is added to filter out outlier designs. To mitigate short-context domination after execution-based filtering, we rebalance the dataset by retaining one-third of short direct-inference datapoints and one-half of short agentic datapoints ($len(\mathcal{R}) \leq 1k$). Stage 1 training incorporates agentic traces generated using the imitation-style model configuration under the worst-state–prioritized synthesis procedure in Section 3.3, together with additional direct-inference samples. Stage 2 training further advances agentic robustness by synthesizing traces using the self-sampling model configuration under the same synthesis algorithm. Each stage is trained by fine-tuning the model checkpoint from the previous stage, starting from the base model.

**Domain-specific syntax constraints.** To increase the yield of executable trajectories during Stage-0 warm-up dataset construction, we augment the teacher's generation prompt with a set of manually curated SystemVerilog validity constraints that target recurrent syntax-level failure modes (e.g., malformed literals, invalid task delimiters, multi-driver assignments). These constraints bias generation toward simulator-compilable testbenches and reduce trivial rejection during trajectory collection. The additional prompt is applied only at generation time for constructing the Stage-0

dataset and is not retained in the stored training samples, ensuring that subsequent training stages operate on standard repository context without specialized prompting. The full rule set is provided in Appendix A.

**Practical intermediate state selection.** For clarity, Algorithm described in Section 3.3 illustrates the worst-state–prioritized procedure using a single selected intermediate state per repository. In practice, to improve data representativeness and stabilize training, we allow selecting up to three intermediate states during synthesis. Specifically, (1) if any intermediate draft results in simulation failure, one such failing state is optionally selected to generate recovery traces targeting compilation or runtime errors; (2) if all sampled states fail simulation, no coverage-ranked worst state exists, therefore only the simulation-failure state is retained; and (3) when successful states exist, an additional median-coverage state is optionally selected if its coverage score differs from the worst state by more than a predefined threshold. This strategy preserves the failure-driven emphasis of worst-state prioritization while preventing over-concentration on a single failure mode under limited simulator budgets.

### 4.3. Main Results

**LLM4COV achieves strong coverage performance despite its small model size.** While direct inference results are included for completeness, multi-turn agentic execution remains substantially more difficult due to compounding errors and state-distribution shift. As shown in Table 2, under both direct inference and single-step agent execution, our 4B-parameter model consistently outperforms hardware-design-specific models (Wei et al., 2025; Zhu et al., 2025; Wang et al., 2025b) and substantially larger models (Meta AI, 2024; Team, 2025; Hui et al., 2024). In particular, nickname achieves 69.2% pass rate plus 90.4% average coverage in CVDP-ECov, and 85.0% pass rate plus 96.3% average coverage in AutoEval-ECov , exceeding the 30B teacher model significantly and matching or surpassing models at the 50×–100× parameter scale. These results demonstrate that effective agentic learning for hardware verification cannot be achieved through model scale alone, but instead requires execution-aware supervision and targeted agentic data synthesis.

### 4.4. Ablation Studies

All evaluations in this subsection are done on CVDP-ECov.

**Student-grounded trajectory synthesis.** Figure 6 examines how different transition-generation strategies behave as the intermediate-state distribution evolves during training. For each variant, we synthesize a dataset using the same simulator budget, dataset size, and intermediate-state selection procedure, and then fine-tune an identical student model. During evaluation, first-round intermediate states are

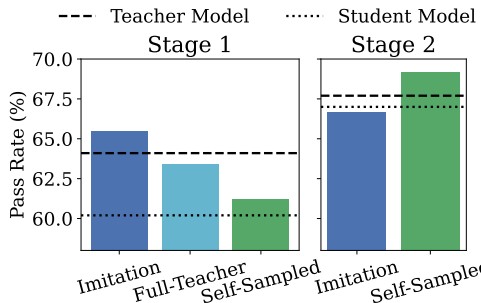

*Figure 6.* Agentic trace taxonomy under intermediate-state distribution drift. Full-teacher traces are omitted in Stage 2 since the relative gap between imitation-style and full-teacher supervision arises from state-distribution mismatch, and is not expected to vary qualitatively with the teacher–student performance gap.

generated once by the stage's student model and kept fixed, isolating the effect of the recovery model used to generate transitions.

In Stage 1, the teacher model substantially outperforms the student. Under this regime, pairing student-induced states with teacher-generated transitions yields stronger learning signals than using teacher-only traces, indicating that aligning supervision with the student-induced state distribution is more important than maintaining a purely teacher-generated trajectory. This observation is consistent with the intuition that corrective supervision should be grounded in the failure modes actually encountered by the student.

In Stage 2, the student approaches the teacher's performance level. We therefore compare imitation-style supervision with fully self-sampling transitions under the same worst-state prioritization strategy. While teacher-guided transitions remain stable, self-sampling transitions become increasingly competitive as the student approaches teacher-level performance. They allow the model to propose repairs that can match or exceed those of a fixed teacher and are naturally aligned with the student's own generation distribution, making them easier to learn from during fine-tuning. Together, these results illustrate a natural progression from teacher-guided correction to student-driven refinement as the intermediate-state distribution shifts during training.

**Worst-State-Prioritized Sampling.** Figure 7 compares intermediate state selection strategies used in Stage 1 training, including *Best-State*, *Uniform*, *Median-State*, and *Worst-State* selection. All strategies share the same intermediate drafts and differ only in how transition synthesis budget is allocated.

To ensure fair comparison, we fix the total simulator call and total SFT data point budget across all strategies. Since worst-state prioritization may select multiple states per repository, we augment other baselines to match the same budget. Specifically, the **Median** strategy additionally selects

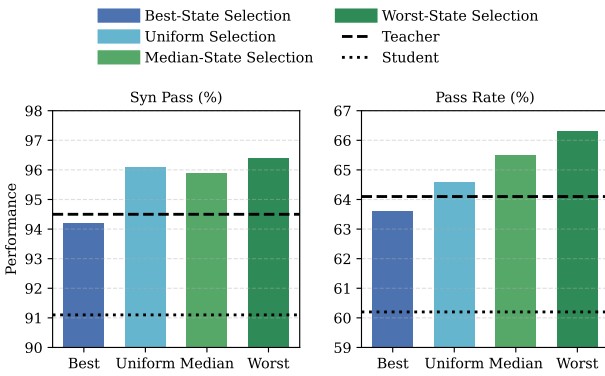

*Figure 7.* Comparison between intermediate state selection strategies in Stage 1. Evaluated under the agentic setting. Syn Pass means percentage of repositories where syntax is correct.

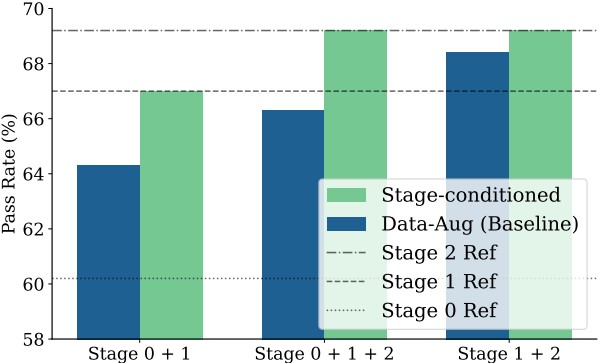

*Figure 8.* Stage-Conditioned Progression, evaluated in agentic setting. Data-Aug stands for naive data augmentation.

*Table 3.* Pass Rate across agentic rounds for teacher, base, and LLM4Cov stages. 30B teacher stands for Qwen3-30B-A3B-2507, and 4B Base stands for Qwen3-4B-2507 (Base).

| | Pass Rate (%) | | | |
|---|---|---|---|---|
| Round Number | $R=0$ | $R=1$ | $R=2$ | $R=3$ |
| 30B Teacher | 29.6% | 49.6% | 58.8% | 63.9% |
| 4B Base | 13.7% | 21.4% | 26.3% | 28.4% |
| Stage 0 | 45.5% | 55.2% | 58.6% | 60.2% |
| Stage 1 | **48.7%** | 59.3% | 63.6% | 67.0% |
| Stage 2 | 48.2% | **62.7%** | **67.5%** | **69.2%** |

the worst state when the coverage gap between median and worst exceeds a threshold. The **Best** strategy selects lower-ranked high-coverage states when budget remains after exhausting the best states. The **Uniform** strategy samples additional intermediate states uniformly when residual budget is available.

Under this controlled setting, performance differences reflect the effect of state prioritization rather than simulator usage. As shown in Figure 7, prioritizing low-coverage intermediate states consistently yields stronger verification performance, validating the effectiveness of worst-state–prioritized supervision.

**Progressive vs. naive data augmentation.** Figure 8 compares stage-conditioned training with naive data augmentation under the same agentic evaluation setting. For the progressive variant, each stage is fine-tuned from the previous checkpoint using its corresponding dataset, preserving alignment between the student model and the distribution of synthesized supervision. For the baseline, we aggregate datasets across stages and train a single model from the same initialization or from a fixed earlier checkpoint.

We observe that stage-conditioned progression consistently outperforms naive augmentation across all dataset combinations. Training on Stage 0+1 and Stage 0+1+2 data sequentially yields higher coverage pass rates than jointly training on the same data from a fixed initialization. Notably, even when using identical Stage 1+2 data, continuing from the Stage 0 checkpoint provides stronger performance than training from scratch on the aggregated dataset. These results indicate that maintaining distributional alignment between the current model and the synthesized supervision is critical for effective learning, and that mixing data from different execution regimes without staging can dilute recovery-focused signals from later stages.

**Sensitivity to Agentic Rounds.** In main evaluation, we set agentic refinement at $N = 3$ rounds; Here we offer an

quantitative ablation on how much does number of agentic rounds affect the final metrics.

Our observation from table 3 is that although all methods improve with more rounds, LLM4Cov consistently achieves higher coverage at every round. Metric gains saturate after 2–3 rounds, indicating most improvements occur early, which justifys our setting of round number at $N = 3$. Such result suggests that the model trained by LLM4Cov framwork can consistently reach strong coverage with fewer simulator calls across different budgets.

## 5. Conclusion

We present LLM4COV, an execution-aware learning framework for high-coverage testbench generation that formulates verification as deterministic, single-step state transitions guided by simulator feedback. By combining execution-validated data curation, worst-state–prioritized synthesis, and progressive agentic supervision, LLM4COV enables models to learn effective verification behaviors directly under agentic execution, where conventional scaling and instruction tuning fall short. Our framework systematically aligns training supervision with simulator-observed failures and coverage bottlenecks, allowing compact models to acquire robust recovery and exploration capabilities that generalize across diverse hardware repositories.

## Impact Statement

This paper presents work whose goal is to advance the field of Machine Learning. There are many potential societal consequences of our work, none of which we feel must be specifically highlighted here.

## Acknowledgment

This research is supported by NVIDIA Academic Grant Program using A100 GPUs on Brev platform.

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

# A. Domain-Specific Syntax Constraints

# B. Experiment Detailed Settings

**SFT Settings.** We trained the model for 1 epoch on each stage, with a learning rate of $1 \times 10^{-5}$ and a batch size of 24. Total context length is adapted with synthetic data to ensure all data are included, peaking at 40,960. All SFT stages are executed on $8 \times$A100 80GB SXM4 GPUs, using 57k data points and approximately 72 GPU hours in total with LlamaFactory. Synthetic data generation takes 420k simulator calls in total.

The following hyperparameters were used during training:

```
learning_rate: 1e-05;
train_batch_size: 1;
eval_batch_size: 8;
seed: 42;
distributed_type: multi-GPU;
num_devices: 4;
```

*Table 4.* Model Evaluation Config. Recommended config is used at best effort.

| Type | Model | Temperature | Top P |
|---|---|---|---|
| General Purpose | llama-4-maverick (400B) | 0.7 | 0.8 |
| | Qwen3-235b-A22b-2507 | 0.7 | 0.8 |
| | Qwen2.5-72B | 0.7 | 0.8 |
| | Qwen3-30B-A3B-2507 | 0.7 | 0.8 |
| Coding Specialized | Qwen2.5-Coder-32B | 0.7 | 0.8 |
| | Qwen3-Coder-30B-A3B | 0.7 | 0.8 |
| | Qwen2.5-Coder-7B | 0.7 | 0.8 |
| Hardware Specific | VeriCoder-Qwen2.5-14B | 0.5 | 0.95 |
| | CodeV-R1-RL-Qwen-7B | 0.6 | 1 |
| | VeriReason-Qwen2.5-7B | 0.5 | 1 |
| LLM4Cov | Qwen3-4B-2507 (Base) | 0.7 | 0.8 |
| | +Stage0 | 0.7 | 0.8 |
| | +Stage1 | 0.7 | 0.8 |
| | +Stage2 | 0.7 | 0.8 |

```
gradient_accumulation_steps: 6;
total_train_batch_size: 24;
total_eval_batch_size: 32;
optimizer: Use OptimizerNames.ADAMW_TORCH with betas=(0.9,0.999) and epsilon=1e-08 and
    optimizer_args=No additional optimizer arguments;
lr_scheduler_type: cosine;
lr_scheduler_warmup_ratio: 0.03;
num_epochs: 1.0;
```

The following packages were used during training:

```
CUDA: 12.4;
NVIDIA Driver: 580.65.06;
llamafactory: 0.9.5;
torch: 2.6.0+cu124;
transformers: 4.57.1;
```

Specifically, in Stage 0 we sampled 87k direct-inference data points and 87k agentic data points with the teacher model (Qwen3-Coder-30B-A3B-Instruct), selected 30k of them (20k direct-inference + 10k agentic), and fine-tuned the base model (Qwen3-4B-Instruct-2507) on them;

In Stage 1 we selected 11k long-context repos among the 87k, and sampled 55k direct-inference data points with the Stage 0 SFT model as the student model; we then applied worst-priority sampling and imitation learning, and collected 67k agentic data points with the teacher model. Finally, we filtered out 8k direct-inference data points and 9k agentic data points, and used them to fine-tune the Stage 0 SFT model;

In Stage 2 we applied a similar method as Stage 1, except using self-sampling to replace imitation learning, and only used agentic data to fine-tune the Stage 1 SFT model.

**EDA Settings.** All hardware simulations and coverage evaluations are performed using Cadence Xcelium and IMC toolchains on a Rocky Linux 8.9 environment. We use xrun (version 22.03-s001) as the SystemVerilog simulator for compilation and execution, and Cadence IMC (version 25.09-a001) for post-simulation coverage analysis.

**Evaluation Settings.** The temperature and top P used by evaluation generation is listed in Table 4.

## C. Execution Validated Dataset

As described in Section 4.2, Stage 0 constructs a warm-up dataset to mitigate the high rate of syntax and execution failures observed in current models. This appendix quantifies the severity of these failures and demonstrates how the proposed warm-up dataset alleviates them in a model-agnostic manner.

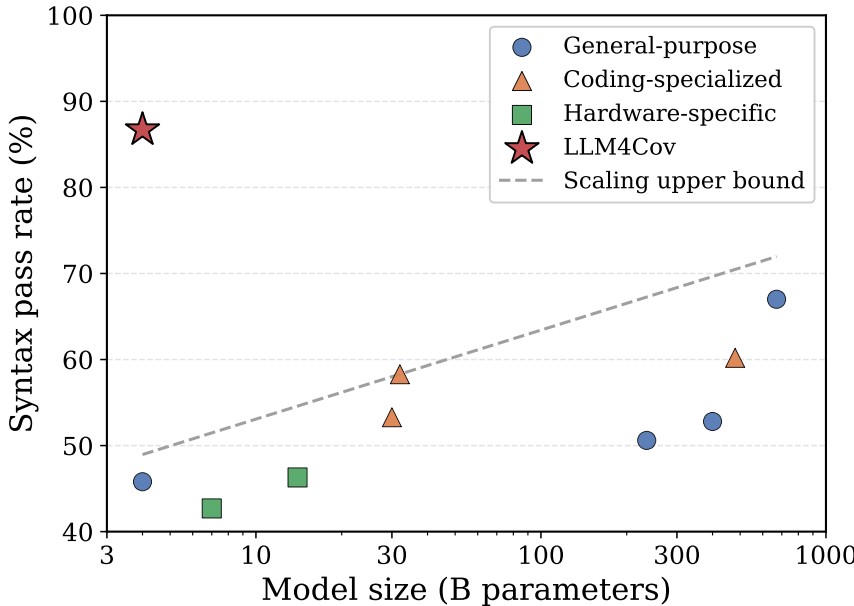

*Figure 9.* Simulator pass rates of existing LLMs on verification stimulus generation, all in instruct mode. Stage 0 model is used here as LLM4Cov. All baseline models has Syn Pass metric below 70%, while our execution-validated dataset enables over 85% pass rate.

*Table 5.* Ablation on execution-based dataset curation. All datasets contain the same number of samples.

| | Syn Pass (%) | |
| --- | --- | --- |
| Type | Direct Infer | Agentic |
| Teacher | 53.3% | 85.1% |
| Teacher with augmented spec | 71.1% | 85.8% |
| Base | 45.8% | 61.0% |
| SFT: Data without filtering | 70.6% | 79.5% |
| SFT: Data with Execution filtering | 83.9% | 87.7% |

### C.1. Syntax Pass Rate of Recent Models

We introduce **Syn Pass** as an additional metric in the CVDP-ECov benchmark: the percentage of repositories for which the generated testbench has correct syntax, compiles successfully and completes simulation. Syn Pass therefore measures compilation and execution validity. Following prior work (Pinckney et al., 2025), assertion generation is treated as a separate task; remaining failures are thus primarily due to syntax errors or simulator timeouts.

Even when restricted to stimulus generation without assertions, current LLMs frequently fail to produce simulator-compilable verification code, as illustrated in Figure 9.

### C.2. Ablation of Proposed Remedies

We apply two techniques when constructing the warm-up dataset:
- **Expert syntax constraints.** As detailed in Appendix A, we introduce a teacher-model-specific rule set into the prompt to prevent common syntax and structural errors.
- **Coverage-based filtering.** We apply coverage-guided rejection (Section 3.3) with a minimum improvement threshold of 1%, together with an additional minimum absolute coverage requirement of 50% to remove outlier designs.

Table 5 analyzes dataset construction choices. Repository specialization substantially reduces teacher-model syntax failures, which explains why SFT on unfiltered data can achieve a higher pass rate than the teacher itself. Execution-based filtering provides the dominant improvement in simulator pass rates under both direct-inference and agentic evaluation.

Figure 10 further shows that fine-tuning on the curated dataset consistently improves syntax correctness across multiple

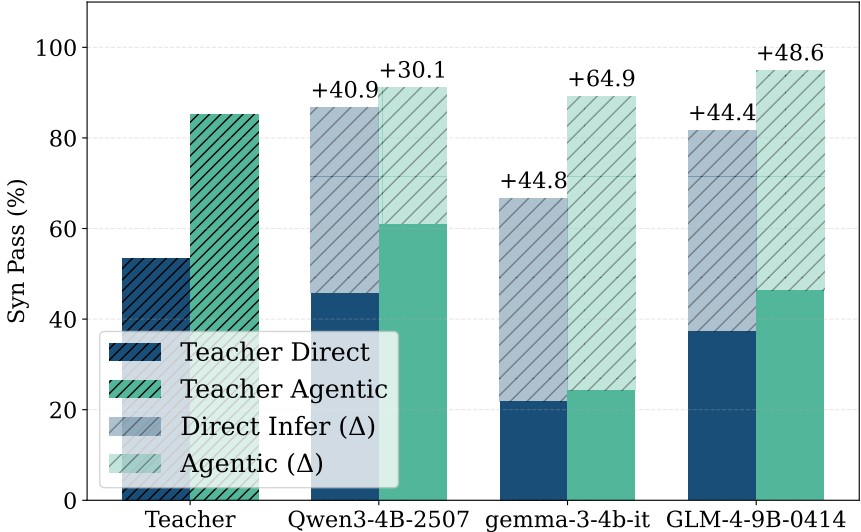

*Figure 10.* Syntax Improvement by Execution-Validated Dataset on models from different families. Fine-tuned on instruct versions.

model families(Team et al., 2025; Team, 2025; Glm et al., 2024), indicating that the execution-validated dataset generalizes beyond a single architecture.

## D. Baseline Comparisons

Direct head-to-head comparison with prior LLM-based testbench generation methods is difficult due to differences in evaluation protocols, toolchains, and model availability. We nonetheless provide a best-effort empirical comparison that shows consistent gains for LLM4Cov across two settings.

**Comparison with TBnTB (as reported).**   TBnTB reports results on a custom protocol using Aldec Riviera-Pro with a 14B model trained on Claude-generated data, whereas LLM4Cov is evaluated on CVDP-ECov with Cadence Xcelium using a 4B student distilled from a 30B teacher. Despite the smaller student model, LLM4Cov achieves substantially higher average coverage, as summarized in Table 6.

*Table 6.* Reported results from TBnTB compared to LLM4Cov. The two methods use different toolchains and evaluation protocols; numbers are not strictly comparable but illustrate a large gap in favor of LLM4Cov.

| Method | Model | Eval setup | Avg Cov% |
|---|---|---|---|
| TBnTB | 14B (Claude-gen. data) | custom, Aldec Riviera-Pro | 25% (reported) |
| LLM4Cov | 4B (30B teacher) | CVDP-ECov, Cadence Xcelium | **90.4%** |

**Comparison with CorrectBench (re-evaluated under AutoEval-ECov).**   We thank the authors of CorrectBench for sharing their generated testbenches. To enable a more controlled comparison, we introduce *AutoEval-ECov*, a coverage-oriented evaluation protocol built on Verilog-Eval, and evaluate both methods under this unified setup. Results are shown in Table 7 which matches numbers in Table 2.

*Table 7.* Re-evaluation of CorrectBench's released testbenches and LLM4Cov under the unified AutoEval-ECov protocol on Verilog-Eval.

| Method | Avg Cov% | Pass Rate (Full Cov Rate) |
|---|---|---|
| CorrectBench (Claude 3.5 Sonnet) | 89.9% | 61.5% |
| LLM4Cov | **96.5%** | **84.0%** |

**Why strict comparison remains limited.** Several factors prevent a fully controlled comparison. First, there is a *protocol mismatch*: LLM4COV is evaluated under CVDP-ECov, where the model has access to the full repository and simulator feedback, whereas CorrectBench and TBnTB are designed primarily for spec-only settings. Second, there is a *toolchain mismatch*: TBnTB reports results using Aldec Riviera-Pro under its own custom setup, while our results are obtained with Cadence Xcelium. Third, *reproducibility and system differences* further complicate comparison: TBnTB does not release model weights, and CorrectBench is a multi-agent framework built on proprietary models, while the evaluated outputs of LLM4COV are fine-tuned open-source models.

**Takeaway.** While not strictly controlled, these comparisons provide consistent evidence that LLM4COV improves functional coverage using a smaller open-source model, with a clear advantage over prior LLM-based approaches.

## E. Limitations and Future Directions

Although this work adopts an offline, execution-grounded supervised learning paradigm, it is not fundamentally incompatible with reasoning-based or reinforcement learning (RL) approaches. Our focus on offline learning is primarily driven by the high cost and limited availability of hardware simulators, where execution is slow and resource-intensive. As shown in Table 8, models across all SFT stages retain substantial output diversity, with consistent Pass@1–Pass@5 gains in both direct and agentic settings, indicating that fine-tuning does not collapse the policy into deterministic behaviors.

*Table 8.* Output diversity check for each SFT stage

| SFT Stage | Direct Pass Rate | | Agentic Pass Rate | |
|---|---|---|---|---|
| | Pass@1 | Pass@5 | Pass@1 | Pass@5 |
| Stage 0 | 45.5% | 68.7% | 60.2% | 77.1% |
| Stage 1 | 48.2% | 68.7% | 67.0% | 84.3% |
| Stage 2 | 50.6% | 72.3% | 69.2% | 81.9% |

These results suggest that execution-grounded SFT functions as a strong and diverse initialization policy rather than a terminal optimization stage. Such diversity-preserving policies are well suited for future RL, where stochasticity is essential for effective exploration and long-horizon credit assignment. Therefore, while offline execution-grounded learning constitutes a complete and effective paradigm in its own right, it can alternatively be viewed as an RL-compatible foundation when online interaction and simulator budgets permit.

