# OpenReview forum: "LLM4Cov: Execution-Aware Agentic Learning for High-coverage Testbench Generation"
_ICML.cc/2026/Conference — ICML 2026 regular_

### Official Review · Reviewer_wrUU · 2026-03-12

**Soundness:** 3
**Presentation:** 3
**Significance:** 3
**Originality:** 3
**Overall Recommendation:** 4
**Confidence:** 4

**Summary:**

This paper proposes an offline execution-grounded learning framework for hardware verification. Their main idea is to model iterative testbench generation as memoryless state transitions that are scored by a simulator, then improve a compact student model through three components: execution-validated data curation, policy-aware agentic trajectory synthesis, and worst-state-prioritized sampling. The evaluation is on a modified benchmark setting, CVDP-ECov, where the full hardware repository is visible to the model; the final 4B model reaches a significant agentic coverage pass, exceeding its 30B teacher and greatly improving over the 4B base model.

**Compliance With Llm Reviewing Policy:**

Affirmed.

**Final Justification:**

Since my initial comments (especially regarding the attribution of gains) were not fully resolved, I keep my initial "weak accept" score.

**Key Questions For Authors:**

**Q1.** How much of the final gain remains if repository specialization rules are removed entirely from stage 0 data generation?


**Q2.** Can the authors provide variance over multiple runs or bootstrap confidence intervals over the 83 repositories?


**Q3.** How sensitive are results to the number of agentic rounds and simulator budget?


**Q4.** Why is "cid012" the only task evaluated? Are there other CVDP tasks where the setup fails or is too expensive?

**Limitations:**

Not fully adequate. The appendix gives some limitations discussion, but the main-paper impact statement is too generic and does not seriously discuss risks, scope limits, or practical constraints.

**Strengths And Weaknesses:**

**Strengths:**

1. The paper targets a practically meaningful domain where execution is slow, non-differentiable, and expensive. This setup seems to have good reasons for doing hardware verification.

2. The three-stage progression (verified curation, imitation-style agentic supervision, self-sampled refinement) is reasonably well formulated. The memoryless formulation, trajectory taxonomy, and worst-state sampling are clear to follow.

3. I think the main table is impressive: the model goes from 28.4% to 69.2% Cov Pass under agentic evaluation, and stage 2 also reaches 97.8% Sim Pass and 90.4% Avg Cov. The ablations support that execution filtering matters, and the plots on page 8 support the claim that worst-state selection and stage-dependent policy choice help.

4. I like that the paper does not only show an end result; it also studies memoryless vs vanilla history, execution filtering, transition-policy choice, and state-selection strategy. These ablations are important.

**Weaknesses:**

1. The evaluation is not on the original CVDP protocol; it is on CVDP-ECov, where the model sees the full repository rather than only the natural-language specification. That may indeed be more practical, but it also makes the task meaningfully different, so the results are not directly comparable to prior benchmark numbers in the original setting. This is probably my biggest concern.

2. The entire study is on a single task (cid012) with 83 repositories. For a methods paper making broad claims about execution-grounded agent learning, I would have liked either more hardware verification tasks, a second benchmark, or at least stronger evidence that the method generalizes beyond this one adapted setting.

3. The paper (on several instances) positions the framework as solving agentic distribution shift offline, but the experimental evidence is still domain-specific and the memoryless assumption only shows a small gain in Table 1. The formulation is plausible here, but I do not think the paper fully establishes it as a generally attractive method beyond this task.

4. The paper compares against large general models and some hardware-specific models, but the strongest contribution is a specialized training pipeline using large-scale synthetic data, simulator filtering, and curriculum design. I would have liked stronger baselines that isolate whether the gains come mainly from the proposed state-distribution ideas versus simply domain-specific execution-filtered SFT at scale. Table 3 helps, but it does not fully show this.

---

> ### Author Rebuttal · Authors · 2026-03-31
>
> We thank the reviewers for their valuable feedback and for recognizing our practical problem setting, clear formulation, strong empirical results, and thorough ablation analysis. The main concerns include **protocol differences, generalization, and empirical robustness (variance, benchmarks, and sensitivity)**, etc. Our responses are provided below:
> # Protocol Change
> We agree that evaluating on CVDP-ECov instead of the original CVDP protocol breaks direct comparability. Our motivation is to enable a **code-reachable** setting where the agent can iteratively interact with RTL and simulator feedback, aligning with practical verification workflows (e.g., unit testing and coverage closure). We therefore view CVDP-ECov not as an arbitrary modification, but as a **necessary adaptation that complements** rather than replaces the original benchmark.
> # Empirical Robustness
> **Metric Variance.** We report **standard errors (SE)** over repeated runs.
>
> Main results (n = 5 runs):
>
> | Method | Cov Pass |
> |-|-|
> | 30B Teacher | 63.9% ± 1.5% |
> | 4B Student Base | 28.4% ± 1.4% |
> | LLM4Cov Stage 0 | 60.2% ± 1.4% |
> | Stage 1 | 67.0% ± 1.3% |
> | Stage 2 | 69.2% ± 1.2% |
>
> To further reduce estimator variance and obtain more stable estimates, we increase the number of repeated runs to n = 40:
>
> Repeated sampling (n = 40 runs):
>
> | Method | Cov Pass |
> |-|-|
> | Stage 1 | 67.35% ± 0.49% |
> | Stage 2 | 69.31% ± 0.48% |
>
> Improvements exceed estimated uncertainty, indicating stability. We will add variance analyses in revision.
>
> **Sensitivity to Agentic Rounds**
> We thank the reviewer for raising this point. We analyze performance as a function of the number of agentic refinement rounds (each round corresponding to additional simulator calls).
>
> | Method | Round 0 | Round 1 | Round 2 | Round 3 |
> |-|-|-|-|-|
> | 30B Teacher | 29.6% | 49.6% | 58.8% | 63.9% |
> | 4B Student Base | 13.7% | 21.4% | 26.3% | 28.4% |
> | LLM4Cov Stage 0 | 45.5% | 55.2% | 58.6% | 60.2% |
> | Stage 1 | 48.7% | 59.3% | 63.6% | 67.0% |
> | Stage 2 | 48.2% | 62.7% | 67.5% | 69.2% |
>
> All methods improve with more rounds, but LLM4Cov **consistently achieves higher coverage** at every round. Gains saturate after 2–3 rounds, indicating most improvements occur early. This suggests LLM4Cov reaches strong coverage with fewer simulator calls.
>
> **Effect of Repo Spec.**
> Repository specialization is used only during Stage-0 data construction to **improve sampling efficiency**—i.e., increasing the fraction of trajectories that pass execution filtering—rather than to directly boost downstream performance. To isolate its effect, we additionally evaluate SFT with data from a spec-free teacher without filtering, which achieves 60.0% Syn Pass. This is substantially below SFT without filtering (70.6%) and SFT with execution filtering (83.9%), while still improving over the base model (45.8%).
> # Additional Benchmark and Task Scope
> We agree that a single benchmark is insufficient to support broad generalization claims. To address this, we add an additional evaluation on **AutoEval-ECov (156 designs from VerilogEval)** and compare against CorrectBench, a multi-agent system based on Claude 3.5 Sonnet. Under the same protocol, LLM4Cov (4B, single-step refinement agent) achieves higher coverage (**96.5% vs. 89.9% Avg Cov**), providing complementary evidence beyond a single setting.
>
> Regarding CVDP, different tasks target distinct verification objectives. The cid012 task is specifically designed for stimulus generation and coverage closure, aligning with our focus, while other tasks (e.g., property generation) are not directly compatible with iterative, coverage-driven evaluation.
> # Generalization of memoryless formulation
> We agree that our empirical results are currently grounded in a specific execution-driven domain; however, we believe this direction is **broadly promising** and increasingly supported by **emerging evidence**. While the gain in Table 1 is modest, the key result is that a **memoryless (history-minimal) state** can maintain competitive performance despite using strictly less information, suggesting that effective agent learning may not require growing interaction histories.
>
> More broadly, this aligns with recent efforts to reduce or restructure history (e.g., Context-Folding; Google’s Chain-of-Agents; Ralph loops). In this sense, our work provides concrete evidence in a realistic setting that such formulations are viable. We will clarify this positioning and refer to our detailed discussion in the response to Reviewer yT1H.
> # Attribution of gains.
> We agree that disentangling the effect of state-distribution design from execution-filtered SFT is valuable. Our current ablations (Table 3 and Figure 5,6) control for execution filtering while varying state selection and transition policy, and show consistent gains from these components. We will further clarify this comparison in the revision to make the attribution more explicit.

---

> > ### Author Rebuttal · Reviewer_wrUU · 2026-04-02
> >
> > Thank you for the detailed rebuttal. However, my main concerns still remain. First, the central evaluation is still conducted in the adapted CVDP-ECov setting, where the full repository is visible to the model, rather than in the original spec-only CVDP protocol. I understand the practical motivation, but I think this remains a meaningfully different task, so the paper’s results are still not directly comparable to prior benchmark numbers in the original setting.
> >
> > Second, although the rebuttal improves the discussion of robustness, but I think the current ablations cannot properly isolate the source of the gains. The paper shows that execution-filtered data curation is important, and it also presents evidence in favor of the later agentic stages, but the current evidence still does not completely separate the effect of the proposed state-distribution ideas from the effect of a strong domain-specific synthetic-data and curriculum pipeline.
> >
> > I would like to maintain my original score.

---

> > > ### Author Response · Authors · 2026-04-06
> > >
> > > We understand that the remaining concerns—particularly regarding the difference in evaluation setting—cannot be fully resolved within the rebuttal scope.
> > >
> > > We are glad that you found the problem setting meaningful, and we appreciate your recognition of the stage-wise design, execution-aware learning framework, and the empirical improvements supported by ablations.
> > >
> > > Thank you again for your time and constructive feedback.

---

### Official Review · Reviewer_yT1H · 2026-03-12

**Soundness:** 3
**Presentation:** 1
**Significance:** 3
**Originality:** 2
**Overall Recommendation:** 4
**Confidence:** 3

**Summary:**

The paper presents LLM4Cov, a framework for fine-tuning LLMs to generate high-coverage testbenches for hardware designs. The work formulates the iterative testbench generation process as a Markovian transition system with feedback from expensive simulator invocations. To address the high cost of interaction with the simulator, the authors employ offline imitation learning, where a smaller LLM is trained using trajectories generated by a larger teacher model.
The method is evaluated on 83 hardware repositories. The results suggest that a Qwen3-4B model trained with LLM4Cov outperforms its teacher model as well as several existing hardware-specific models.

**Compliance With Llm Reviewing Policy:**

Affirmed.

**Final Justification:**

The authors’ responses address my concerns, and I believe the manuscript will be improved after incorporating the rebuttal, particularly in terms of presentation. Therefore, I recommend a weak accept.

**Key Questions For Authors:**

See Weakness.

**Limitations:**

yes

**Strengths And Weaknesses:**

## Strength

- The paper addresses an important and challenging problem. Generating high-quality hardware testbenches with high coverage is difficult, especially when feedback requires expensive simulator calls.
- The authors propose combining several techniques, including synthetic data generation, offline imitation learning, and teacher-student training, to tackle this problem. The experimental results show that the trained student model can outperform the larger teacher model and other baselines. The ablation studies suggest that the individual components of the framework each contribute to the final performance.

## Weakness

- The main contribution of the paper is not sufficiently clear. It is unclear whether the primary contribution is a new framework for offline agent learning, or an application of existing techniques for the specific hardware problem. In either case, the current state of the art or common practice in offline agent learning is not clearly described in the introduction or background. As a result, it is difficult to assess which aspects of the proposed system are novel.

- Although Figure 3 provides a high-level architecture overview, the workflow and the relationships between the different components are not sufficiently clear, such as where trajectory synthesis and sampling fit in the whole system. I suggest presenting a clearer overview of the full pipeline first, and then explaining how each component fits into the overall framework. In addition, the motivation for the offline imitation learning setup could be better justified, for example, why the choice to train a student model from a teacher model.

- The paper repeatedly emphasizes the “memoryless” modeling. First, the modeling itself should be introduced earlier in the paper, since the reader encounters this "memoryless" term without sufficient context. Second, the Markovian modeling itself appears fairly standard, since most RL settings rely on the Markov property, and non-Markovian problems can be easily converted into Markovian formulations. I would appreciate a clearer explanation of how the authors’ use of this modeling is novel.

- The baselines, especially the "Hardware Design-Specific" ones, are not well explained.

---

> ### Author Rebuttal · Authors · 2026-03-31
>
> We thank the reviewer for providing the valuable feedback. The main concerns include the contribution of **memoryless settings**, our work’s position on existent **Offline Agentic Learning** research, **clearer presentations** and **baseline explanations**. Our responses are provided below:
> # Explanation on Memoryless/Markov Modeling
>
> We agree that Markov formulations are standard in RL, and that non-Markovian problems can be made Markovian by including full history (e.g., Agent-R1). Our contribution is **not this standard construction**.
>
> Instead, we ask whether agent learning can operate with a **history-minimal state**. Concretely, we retain only the current codebase, latest execution feedback, and the most recent interaction, rather than the full trajectory. While more restrictive, we show this is sufficient for strong performance.
>
> We believe this **direction** is important. Prior agents typically rely on growing interaction histories (e.g., ReAct), yet recent work and reports highlight degradation in long-context settings (“context rot,” Chroma Research) and motivate reducing or restructuring history via compression or decomposition (e.g., Context-Folding; Google’s Chain-of-Agents; Ralph loops).
>
> Our approach can be viewed as a **principled extreme** of this trend: instead of compressing history, we minimize it and rely on the current execution state. We do not claim novelty in the Markov assumption itself, but in demonstrating that in our scenario, this **history-minimal state abstraction is effective** for agent learning.
>
> # Current State on Offline Agentic Learning
> We thank the reviewer for raising this concern and agree that the distinction between general offline agent learning and our specific contributions should be made clearer.
>
> Prior work on offline LLM agent learning has primarily focused on (i) demonstrating feasibility of trajectory-level SFT at scale (AgentBank, EMNLP’24), (ii) assigning value to intermediate steps or actions in long-horizon agent trajectories (AgentPro, EMNLP’25;QLASS, ICML’25), and (iii) scoring entire trajectories for data selection (EDGE, IJCAI’25).
>
> In contrast, our work targets a **different and underexplored regime**: **single-step, iterative refinement agents with existent dense, ground-truth scoring (coverage) for each step**. This setting departs from multi-step credit assignment and instead raises a distinct optimization problem—how to construct and select training data when each transition is directly verifiable and comparable. Our contributions—**Policy-Aware Agentic Data Synthesis** and **Worst-State-Prioritized Sampling**—are designed specifically for this regime, serving as an execution-grounded agent learning framework enabling effective offline learning.
>
> We will revise the Related Work part to more explicitly position our work within this landscape and clarify these distinctions.
>
> # Overview Clarity
>
> We thank the reviewer for pointing out this clarity issue. We note that some reviewers were able to follow this part, suggesting the issue may relate to insufficient emphasis rather than missing content. In the revision, we will make the workflow around Figure 3 more explicit by adding a concise pipeline overview and clarifying where trajectory synthesis, sampling, and stage-wise training fit within the framework. We will also refine Figure 3 to better illustrate these relationships and improve readability, and briefly clarify the motivation for our offline imitation learning setup.
>
> # Hardware Design-Specific Baseline
>
> We thank the reviewer for pointing this out and agree that the “hardware design–specific” baselines may be less familiar to an ML audience. In particular, **hardware design** focuses on generating or completing RTL implementations from specifications, while our task is **hardware verification**, which instead requires constructing testbenches that interact with a fixed design to expose corner cases and maximize coverage. As a result, design-oriented methods are **not directly optimized** for feedback-driven stimulus generation or coverage improvement, which explains their relatively weaker performance on our task.  We will clarify this distinction in the revision.

---

> > ### Author Rebuttal · Reviewer_yT1H · 2026-03-31
> >
> > I thank the authors for their response. I would also like to apologize, as when I submitted my initial review, I mistakenly failed to include the “Strengths” section from my editor. I am now including it.
> >
> > I am satisfied with the rebuttal, especially the clarification of the modeling contributions and how the authors’ work differentiates itself from prior work on offline agent learning. I strongly agree with reviewer XCC2’s comments regarding “over-formalization and mathematical obfuscation” and “terminology and theoretical framing” in the original version. However, I believe the authors will be able to address these concerns in the revision.
> >
> > Overall, I expect the paper’s quality to improve after incorporating the reviewers’ feedback. Therefore, I am raising my score.

---

> > > ### Author Response · Authors · 2026-04-06
> > >
> > > Thanks for the reviewer's suggestion in the whole process! We’re glad the clarifications and the planned modifications addressed your concerns, and we appreciate your note about increasing your score.

---

### Official Review · Reviewer_XCC2 · 2026-03-12

**Soundness:** 3
**Presentation:** 1
**Significance:** 2
**Originality:** 3
**Overall Recommendation:** 4
**Confidence:** 4

**Summary:**

The paper proposes a novel, execution-grounded fine-tuning methodology and benchmark for LLM-assisted hardware verification. Specifically, the framework focuses on automated stimulus generation, utilizing iterative feedback from industry-standard EDA simulators to maximize coverage metrics. To circumvent the high computational costs of online reinforcement learning, the authors employ an offline data synthesis approach, distilling the iterative debugging trajectories of a larger teacher model into a specialized 4B-parameter agent.

**Compliance With Llm Reviewing Policy:**

Affirmed.

**Final Justification:**

The author's rebuttal clarified my misunderstandings.

**Key Questions For Authors:**

1. **Mathematical Formalism:** Could the authors justify the heavy reliance on complex mathematical formalism and dense theoretical terminology (e.g., "memoryless state transitions") to describe standard heuristic debugging loops? Would the authors consider simplifying the notation and language to improve the paper's readability and practical accessibility, or at least providing a brief explanation for them?
2. **Empirical Comparison to Related Works:** Given that functionally similar, execution-grounded methodologies (such as *CorrectBench* and *TB or Not TB*) are cited, what is the justification for excluding a quantitative or empirical comparison against these frameworks? How does LLM4Cov definitively improve upon them?
3. **Fuzzing Baselines:** What is the theoretical or practical rationale for excluding a Constrained Random Verification (CRV) or standard coverage-guided fuzzing baseline to properly contextualize the reported 69.2% coverage metric?
4. **Incremental Coverage Proof:** Without an incremental coverage analysis (e.g., evaluating the LLM's test vectors on top of a baseline pool of random vectors), how does the current methodology mathematically or empirically rule out the generation of trivial, random-equivalent test cases?
5. **Frontier Model Performance:** How does the specialized 4B-parameter model perform when benchmarked against out-of-the-box, state-of-the-art frontier models (e.g., Claude 3.5 Sonnet, GPT-4o) using an identical simulator feedback loop?
6. **Enterprise Applicability:** In real-world enterprise environments where compute budgets are substantial and tape-out risks are extraordinarily high, what is the practical incentive for deploying a specialized 4B model instead of utilizing massively parameterized models (>100B) to maximize verification assurance?

**Limitations:**

Yes

**Strengths And Weaknesses:**

### Methodological Strengths and Contributions
* **Precise Scoping of Stimulus Generation:** The manuscript is commendable for clearly defining its scope. By isolating the generation of input patterns (stimuli) for coverage maximization, the authors successfully decouple this task from the orthogonal and highly complex challenge of assertion generation.
* **Identification of Practical Verification Bottlenecks:** The authors demonstrate a strong understanding of current industry challenges. They rightly identify that achieving coverage closure requires resolving syntax errors and deep coverage holes—tasks that traditional Electronic Design Automation (EDA) tools can diagnose but cannot autonomously rectify.
* **Automation of the Iterative Feedback Loop:** Coupling the Large Language Model (LLM) directly with industry-standard EDA tools (e.g., Cadence Xcelium) is a pragmatic and highly effective approach. This framework successfully automates the manual, iterative engineering labor required to parse simulator logs and refine testbenches.

### Methodological Limitations and Areas for Improvement
* **Over-Formalization and Mathematical Obfuscation:** The manuscript heavily relies on formal mathematical notation to describe straightforward software engineering workflows. Without adequate accompanying plaintext explanations, this over-formalization does more harm than good, forcing the reader to expend unnecessary mental energy decoding trivial concepts dressed in complex notation.
* **Terminology and Theoretical Framing:** The text frequently obscures practical engineering loops behind dense theoretical terminology. Describing the process as "memoryless state transitions guided by deterministic evaluators" complicates what is functionally a heuristic auto-grader and test-driven development loop.
* **Superficial Treatment of Related Works:** The paper briefly cites highly relevant, functionally similar execution-grounded frameworks (e.g., *CorrectBench*, *TB or Not TB*) but dismisses them without conducting any empirical comparisons. Failing to benchmark against these closely related methods makes it impossible to ascertain the true methodological advantages of LLM4Cov.
* **Absence of Classical Verification Baselines:** The experimental design omits comparisons to standard classical verification methodologies. Without a Constrained Random Verification (CRV) or traditional coverage-guided fuzzing baseline, it is difficult to quantitatively assess the framework's true efficiency gains over existing industry workflows.
* **Lack of Incremental Coverage Analysis:** The study reports absolute coverage pass rates (69.2%) without providing an incremental delta over a robust random baseline. It is necessary to evaluate the LLM-generated vectors in conjunction with a large pool of random vectors to prove the model is targeting complex corner cases rather than generating trivial tests.
* **Exclusion of State-of-the-Art (SOTA) Baselines:** The evaluation is restricted to open-weight models (e.g., Qwen, Llama), completely omitting comparisons with highly capable proprietary frontier models (e.g., Claude 3.5 Sonnet, GPT-4o). This leaves a critical gap in demonstrating whether the specialized training framework provides a tangible advantage over current frontier APIs.
* **Practical Applicability and Model Constraints:** The stringent focus on a computationally constrained 4B-parameter model lacks alignment with the realities of enterprise-level hardware verification. Given the substantial financial risks associated with chip tape-outs, industrial practitioners typically prioritize the higher accuracy of massive (>100B) models over the inference efficiency of smaller models.

---

> ### Author Rebuttal · Authors · 2026-03-31
>
> We thank the reviewers for their valuable feedback and for recognizing our clear scoping, bottleneck identification and industry alignment. The main concerns include Contextualization of Cov Pass Metric, Empirical Comparison to Related Works, Enterprise Applicability, and Mathematical Formalism. Our responses are provided below:
> # Contextualization of Cov Pass Metric
> We thank the reviewer for raising the concern regarding the lack of context for the reported coverage metric and its relation to industrial workflows.
>
> We would like to clarify that the reported **69.2%** is not the **average coverage** percentage, but the **coverage pass rate**—i.e., the fraction of designs that meet predefined coverage thresholds. These thresholds are specified by experienced hardware verification engineers in the benchmark and are intended to reflect practical coverage closure criteria on a per-design basis. In contrast, the corresponding average coverage achieved by our method is **90.4%**, which better reflects the absolute coverage level.
>
> We agree that the current terminology (“Cov Pass”) may be confusing, and will revise it to clearer naming such as **“Pass Rate”** and explicitly distinguish it from average coverage (%) in both text and figures to improve interpretability. We also agree that comparisons to classical baselines such as CRV or coverage-guided fuzzing would provide additional context, and consider this an important direction for future evaluation.
>
> # Empirical Comparison to Related Works
> We thank the reviewer for highlighting the importance of empirical comparison to related execution-grounded frameworks.
>
> **Best-effort empirical comparison.**
> We provide approximate comparisons showing consistent gains.
>
> **(1) TBnTB (reported results vs. LLM4Cov)**
>
> | Method | Model | Eval setup | Avg Cov% |
> |-|-|-|-|
> | TBnTB | 14B (Claude-generated data) | custom protocol, Aldec Riviera-Pro | 25% (reported)  |
> | LLM4Cov | 4B (30B teacher) | CVDP-ECov, Cadence Xcelium | **90.4%** |
>
> **(2) CorrectBench (re-evaluated under AutoEval-ECov)**
> We thank the authors for sharing their generated testbenches. We introduce **AutoEval-ECov**, a coverage-oriented protocol on Verilog-Eval, and evaluate both methods under this unified setup.
>
> | Method | Avg Cov% | Full Cov Rate |
> |-|-|-|
> | CorrectBench (Claude 3.5 Sonnet) | 89.9% | 61.5% |
> | LLM4Cov | **96.5%** | **84.0%** |
>
> **Why strict comparison remains limited.**
> - **Protocol mismatch.** LLM4Cov is evaluated under **CVDP-ECov**, where the model has access to the full repository and simulator feedback; CorrectBench and TBnTB are designed primarily for spec-only settings.
> - **Toolchain mismatch.** TBnTB reports results under its own custom setup using **Aldec Riviera-Pro**, whereas our results are obtained under **Cadence Xcelium**.
> - **Reproducibility and system differences.** TBnTB does not release model weights; CorrectBench is a multi-agent framework evaluated with proprietary models, while LLM4Cov evaluates a single fine-tuned open-source model.
>
> **Takeaway.**
> While not strictly controlled, these results provide consistent evidence that LLM4Cov improves coverage with a **smaller open-source model**, with advantage over related approaches.
> # Enterprise Applicability and Frontier Model
>
> We agree that in real enterprise tape-out workflows, the objective is verification assurance rather than minimizing model size, and in such settings practitioners may well prefer to use very large models if cost is justified.
>
> Our goal in studying a 4B model is different: we use it as a constrained but informative testbed to evaluate whether LLM4Cov provides a fundamentally stronger learning signal. The main takeaway is not “a 4B model is enough for enterprise deployment,” but rather “our framework can dramatically amplify the capability of an open model, even under limited scale.” This makes the result scientifically useful: demonstrating strong gains in the low-resource regime is evidence that the method may **transfer even more effectively to larger open models** when additional training budgets are available.
>
> # Mathematical Formalism and Terminalogy
> We thank the reviewer for highlighting this gap. We agree that the framework can be naturally understood as an iterative, feedback-driven refinement loop, and that the current formalism may obscure this intuition.
>
> Our intent was to provide a precise description of the single-step refinement setting and distinguish it from history-dependent multi-step agents. However, we acknowledge that this level of formalism is not necessary for all readers and can reduce readability.
>
> In the revision, we will: (1) introduce an intuitive, plain-language description before formalization, (2) move detailed notation to the appendix, (3) simplify terminology (e.g., replacing “memoryless state transitions” with “single-step refinement based on current state”), and (4) improve figures to better illustrate the iterative loop.

---

> > ### Author Rebuttal · Reviewer_XCC2 · 2026-03-31
> >
> > I thank the authors for their rebuttal. My questions are addressed appropriately, and I am increasing my score.

---

> > > ### Author Response · Authors · 2026-04-06
> > >
> > > Thanks for the reviewer's suggestion in the whole process! We’re glad the clarifications addressed your concerns, and we appreciate your note about increasing your score.

---

### Official Review · Reviewer_bWAn · 2026-03-15

**Soundness:** 2
**Presentation:** 2
**Significance:** 2
**Originality:** 2
**Overall Recommendation:** 2
**Confidence:** 3

**Summary:**

This paper studies hardware verification in a setting where simulator feedback is reliable but expensive, and where strong one-shot code generation does not naturally transfer to strong multi-turn agent behavior. To address this, the paper proposes an offline training pipeline that combines execution-validated data curation, synthetic agent trajectories, and self-sampling refinement to train a compact verification agent without relying on online RL.

**Compliance With Llm Reviewing Policy:**

Affirmed.

**Final Justification:**

Remain my original evaluation as no further empirical evidence was provided during rebuttal stage.

**Key Questions For Authors:**

See above.

**Limitations:**

See above.

**Strengths And Weaknesses:**

## Strengths

* The paper tackles a practically important problem. Hardware verification is a meaningful setting where execution feedback is reliable but expensive, and where direct one-shot generation does not naturally translate into strong multi-turn agent behavior.

* The overall training recipe is coherent. The Stage 0 / Stage 1 / Stage 2 decomposition is easy to follow, and the motivation for each stage is sensible: execution-validated data curation, imitation-style learning to address student-teacher mismatch, and self-sampled training once the student becomes stronger.

* The empirical results are strong within the benchmark. In particular, the final 4B model performs very well in the agentic setting and reportedly surpasses the much larger teacher on Cov Pass, which is a notable result.

## Weaknesses

* The main contribution feels more like a strong domain-specific training pipeline than a sharp new algorithmic idea. Many of the ingredients are intuitive and sensible, but the paper is less clear on what the central methodological novelty really is beyond the overall recipe.

* The motivation for avoiding online RL is reasonable, given the paper’s emphasis on slow hardware simulation and limited simulator budgets. However, I still found it somewhat unclear, in a more quantitative sense, how prohibitive online methods would actually be in this setting.

* It is also somewhat hard to isolate where the gains are coming from. Stage 0 data curation already seems quite strong, so part of the final improvement may come from better execution-validated training data rather than from the later agent-learning stages themselves.

* More broadly, the hardware verification setting is important, but also quite specialized. As written, the paper comes across more as a strong domain-specific training pipeline than a broadly impactful ML contribution for the ICML community, and may be better suited to a more application- or EDA-oriented venue.


Overall, I think the problem is meaningful, the method is sensible, and the experimental results are strong within the chosen setting. At the same time, I am less convinced by the methodological novelty, and I am not sure the contribution is broad enough to strongly resonate with the core ICML audience.

---

> ### Author Rebuttal · Authors · 2026-03-31
>
> We thank the reviewer for their thoughtful feedback and for recognizing the **practical importance**, **coherent stage-wise design**, and **strong empirical performance** of our approach. We agree that, as written, the work may appear as a domain-specific training pipeline; our intent, however, is to highlight a more general **agent learning abstraction under dense, verifiable feedback**.
>
> **On methodological novelty.** Our contribution is not the individual components (e.g., SFT, filtering), but identifying and formalizing an underexplored regime: *single-step, iterative refinement agents with dense ground-truth evaluation per step*. Unlike prior offline agent learning (trajectory scoring, credit assignment), this setting enables direct comparison between candidate transitions. Our framework—**worst-state-prioritized sampling** and **policy-aware agentic data synthesis**—is specifically designed for this regime and provides a principled way to construct supervision from execution feedback.
>
> **On memoryless (history-minimal) modeling.** While Markov assumptions are standard, we show that a *history-minimal state* (current code + latest execution feedback) is sufficient for effective agent learning, avoiding reliance on growing interaction histories. This connects to a broader trend in LLM agents toward reducing long-context dependence, and we view our result as a concrete demonstration of this direction under reliable evaluators.
>
> **On isolating gains.** We agree Stage 0 is strong; our ablations show consistent improvements from Stage 1 → Stage 2 (e.g., 60.2% → 67.0% → 69.2% Cov Pass), indicating that later agent-learning stages contribute beyond data curation alone. We will clarify this decomposition more explicitly.
>
> **On online RL feasibility.** In our setting, each simulator call can take seconds (academic) to minutes or hours (industrial). This makes standard online RL (requiring repeated environment interaction and exploration) practically prohibitive under realistic budgets, motivating our offline formulation.
>
> Finally, while we agree hardware verification is a specialized domain, we believe the **underlying abstraction—agentic generation with dense, verifiable feedback—extends beyond this setting**, and we will revise the paper to better emphasize this broader perspective.

---

> > ### Author Rebuttal · Reviewer_bWAn · 2026-04-03
> >
> > Thank you for the response. I am not convinced that combining existing ideas (e.g., execution-guided code generation, DAgger, filtering) constitutes a new paradigm. There are combinatorial ways to compose prior methods, and the authors themselves note that the components are not novel.
> >
> > As suggested by the authors, simulator calls in academic settings are on the order of seconds and may be manageable with sufficient parallelism. Given that 420k simulator calls are used for data construction, it seems reasonable to consider small-scale RL (e.g., ~10k calls per epoch, roughly ~8 CPU hours on a single core assuming ~3 seconds per call, which can be further amortized through parallelism).
> >
> > It remains unclear to me whether the additional gains from Stage 0 → Stage 2 (60.2% → 67.0% → 69.2%) justify the increased engineering complexity, as it is unclear whether a modest RL approach (potentially combined with similar filtering) could achieve greater improvements under a managble interaction budget.
> >
> > I will therefore keep my original evaluation.

---

> > > ### Author Response · Authors · 2026-04-06
> > >
> > > Thank you for the thoughtful follow-up and for engaging deeply with our work. We understand your concerns regarding the level of methodological novelty and the comparison to potential RL-based approaches, and recognize that these aspects cannot be fully addressed within the rebuttal scope.
> > >
> > > We are glad that you found the problem setting meaningful, the training formulation coherent, and the empirical results strong within the benchmark. We also appreciate your perspective on alternative approaches, which we will take into account to further clarify positioning and comparisons in future revisions.
> > >
> > > Thank you again for your time and constructive feedback.

---

### Decision · Program_Chairs · 2026-04-30

**Decision:**

Accept (regular)

**Comment:**

Reviewers appreciated that this work tackles an important and challenging problem, that the method consists of clear and compelling innovations, that the presentation was mostly very clear, and that the empirical results show clear performance benefits and insightful ablations. Many reviewer concerns have been sufficiently resolved, including lack of clarity on the contributions and their novelty, over-formalization of the mathematical and technical presentation, and insufficient clarity of the workflow. The two central remaining concerns are regarding the separation of the precise source of performance benefits and the broadness of applicability.

These remaining concerns are true to some extent of essentially every innovative paper, so there's something of a judgement call in how concerning they really are in this case. I don't find them to merit rejection, but can only weakly recommend acceptance.